# Family Meals and Social Eating Behavior and Their Association with Disordered Eating among Spanish Adolescents: The EHDLA Study

**DOI:** 10.3390/nu16070951

**Published:** 2024-03-26

**Authors:** José Francisco López-Gil, Desirée Victoria-Montesinos, Héctor Gutiérrez-Espinoza, Estela Jiménez-López

**Affiliations:** 1Department of Communication and Education, Universidad Loyola Andalucía, 41704 Seville, Spain; josefranciscolopezgil@gmail.com; 2One Health Research Group, Universidad de Las Américas, Quito 170124, Ecuador; 3Faculty of Pharmacy and Nutrition, Universidad Católica San Antonio de Murcia, 30830 Murcia, Spain; dvictoria@ucam.edu; 4Health and Social Research Center, Universidad de Castilla-La Mancha, 16002 Cuenca, Spain; estela.jimenezlopez@uclm.es; 5Centro de Investigación Biomédica en Red de Salud Mental (CIBERSAM), Instituto de Salud Carlos III, 28029 Madrid, Spain

**Keywords:** eating disorders, eating habits, prevention, family, youth

## Abstract

Purpose: The aim of this study was to examine the association of family meals and social eating behavior with disordered eating behavior in Spanish adolescents. Methods: This was a cross-sectional study that included 706 adolescents (43.9% boys) from the Eating Habits and Daily Life Activities (EHDLA) study (aged 12 to 17) from *Valle de Ricote*, Region of Murcia, Spain. The frequency of family meals was assessed by asking participants to report how often their family, or most household members, had shared meals in the past week. Social eating behavior was evaluated using three statements: “I enjoy sitting down with family or friends for a meal”, “Having at least one meal a day with others (family or friends) is important to me”, and “I usually have dinner with others”. To evaluate disordered eating, two psychologists administered the Sick, Control, One, Fat and Food (SCOFF) questionnaire. Results: After adjusting for several covariates, for each additional family meal, the likelihood of having disordered eating behavior was lower (odds ratio (OR) = 0.96; 95% confidence interval (CI) 0.93 to 0.9997, *p* = 0.049). On the other hand, a lower likelihood of having disordered eating behavior was observed for each additional point in the social eating behavior scale (OR = 0.85; 95% CI 0.77 to 0.93, *p* = 0.001). The likelihood of having disordered eating behavior was 0.7% lower for each additional family meal (95% CI 0.01% to 1.4%, *p* = 0.046). Furthermore, for each additional point in the social eating behavior scale, a lower probability of having disordered eating behavior was observed (3.2%; 95% CI 1.4% to 5.0%, *p* < 0.001). Conclusions: While disordered eating behavior is complex and can be shaped by various factors, both family meals and social eating behavior emerge as significant factors inversely associated with this condition among adolescents. Promoting regular engagement in family meals and fostering positive social eating experiences could serve as effective strategies in public health initiatives aimed at mitigating the incidence of disordered eating behavior among the young population.

## 1. Introduction

Eating disorders are complex mental health conditions characterized by abnormal eating or weight control behaviors, which lead to health or psychosocial impairments [1]. Eating disorders are a significant threat to health since these are associated with serious medical complications [2] and have been considered the second, among mental disorders (following substance use disorders), associated with a substantial reduction in life expectancy, with 16.64 years of potential life lost [3]. These disorders are particularly prevalent and may be especially deleterious [4] during adolescence [5], a critical developmental stage marked by significant physical, cognitive, and socio-affective changes [2]. In this sense, the importance of eating disorders among adolescents as a public health concern has been emphasized by several studies [2,6,7,8].

Even in the absence of meeting full diagnostic criteria, disordered eating behaviors, which refer to a variety of behaviors such as limiting food intake, eating in excess, practicing excessive exercise, self-induced vomiting, and using laxatives or diuretics [9], are also a public health issue, firstly, due to their prevalence in children and adolescents [10]. Furthermore, although not categorized as eating disorders, these have been associated with negative physical, psychological, and social consequences [11]. Thus, disordered eating behavior is related to an increased risk of developing eating disorders [12] and obesity later in life [13], higher symptoms of depression [14], higher risk of alcohol, tobacco, and central nervous stimulants consumption [15], reduced wellbeing [15], and higher risk for reaching lower socioeconomic achievement [16]. Therefore, it seems crucial to differentiate and examine disordered eating behavior and eating disorders separately [17] to promote early detection. In addition, considering the suboptimal rates of remission linked to current evidence-based therapies for eating disorders [18,19], a thorough exploration and understanding of the factors related to these conditions may be of relevance in order to develop detailed interventions aimed at prevention and optimal treatment approaches.

Social eating behavior stands out as an environmental factor with potential implications for the onset and progression of eating disorders in adolescence [20,21]. In this sense, social eating behavior is a complex interplay of physiological, psychological, and social factors that influence meal timing, food quantity, and food choices [22]. Eating habits are highly influenced by the social environment (i.e., family, friends, and media) [23]. Variations in our food consumption can occur when eating is conducted in the company of others or when we are alone, as our dietary selections mirror those of our close social circles [22]. Furthermore, social eating behavior includes the various feelings and emotions that occur when eating with others, like family and friends, and can influence one’s relationship with food and body image [24]. Therefore, this social eating behavior can be an important factor in molding eating behavior in adolescents, creating specific relationships and feelings towards food and body image [25,26].

On the other hand, family meals, which reflect family dynamics and are often influenced by family and societal norms, have a significant role in shaping adolescents’ views and habits related to food [27]. These family meals can be determinant for building social connections and enhancing relationships [28,29]. However, the potential benefits obtained by family meals on social relationships and eating habits among adolescents could depend on the perception of the situation, since these are not always enjoyed. Thus, previous findings have underlined the importance of voluntary participation in family meals for developing healthy eating habits and strengthening family ties, pointing out its preventive role against eating disorders and other health issues [30,31]. When participation is forced, it may not have the intended impact and could even be harmful, possibly reducing the potential of these interactions to encourage healthy eating [26,28].

Several studies have explored various aspects related to eating disorders, social eating behavior, and family meals in adolescents [30,32,33]. However, the comprehensive exploration of the diverse environmental factors contributing to disordered eating behavior is still a pressing need [26,34]. Advances in research have highlighted the role of various social determinants, including economic, environmental, and cultural domains, in the understanding of mental health conditions [35,36,37]. Moreover, exploring the attitudes and actions of adolescents regarding meal organization and communal dining, as well as examining the connections with their experiences of early family meals may be of relevance [38].

Therefore, the aim of this study was to examine the association of family meals and social eating behavior with disordered eating behavior in Spanish adolescents, a perspective that, to our knowledge, has not been previously explored. We hypothesize that adolescents who eat a greater number of meals or have higher social eating behavior will be less likely to have disordered eating behavior. This effort could be relevant, especially considering the high prevalence of disordered eating behavior among adolescents [10] and the role of family and social environments on eating behaviors in this population [39]. By exploring the complex connections between these important factors, this study seeks to expand the current body of knowledge and set the foundation for developing new intervention strategies to reduce the associated risks in the adolescent population.

## 2. Material and Methods

In this study, we used data from the Eating Habits and Daily Life Activities (EHDLA) study, following the methodology described by López-Gil [40]. This secondary cross-sectional study included 706 adolescents (43.9% boys). The EHDLA study focused on adolescents aged 12 to 17 from *Valle de Ricote*, Region of Murcia, Spain, and took place in all three secondary schools during the 2021/2022 academic year.

Before the adolescents joined the study, their parents or guardians were required to sign an informed consent form. Both the adolescents and their legal guardians were provided with a document that explained the study’s objectives, the activities involved, and the questionnaires that would be used. The adolescents were also asked to agree to participate.

The eligibility criteria for the study were limited to adolescents aged 12 to 17 who were either residents or registered in *Valle de Ricote*. The study excluded any students who were exempt from secondary school physical education, as the testing and questionnaire administration occurred during those lessons. Furthermore, individuals with specific pathologies requiring special attention, those undergoing pharmacological treatment, those who did not provide consent for participation, or those whose parents or legal guardians did not grant permission were not included in the study.

The study received ethical approval from the Bioethics Committee of the University of Murcia and the Ethics Committee of the Albacete University Hospital Complex (approval IDs 2218/2018 and 2021–85, respectively). The study was carried out in compliance with the Helsinki Declaration to ensure the protection of the participants’ rights.

### 2.1. Measurements

#### 2.1.1. Family Meals (Independent Variable)

We assessed the frequency of family meals by asking participants to report how often their family, or most household members, had shared meals in the past week. They responded using an ordinal scale with options from (a) “none” to (h) “seven days”. We collected this data separately for breakfast, lunch, and dinner. To determine the total frequency of family meals for the week, we summed the reported instances for each meal [30].

#### 2.1.2. Social Eating Behavior (Independent Variable)

We measured social eating behavior using three statements: “I enjoy sitting down with family or friends for a meal”, “Having at least one meal a day with others (family or friends) is important to me”, and “I usually have dinner with others”. Participants could respond with “strongly disagree”, “somewhat disagree”, “somewhat agree”, or “strongly agree” for each statement. These responses were scored with four, three, two, and one point, respectively. The scores obtained were summed to determine the social eating behavior, which could range from 3 to 12 points. A higher score indicates more frequent social eating behavior. The reliability of these items, with a Cronbach’s alpha of 0.70, has been established in Project EAT (Eating and Activity over Time) [38].

#### 2.1.3. Disordered Eating (Dependent Variable)

To evaluate disordered eating, two psychologists administered the Sick, Control, One, Fat and Food (SCOFF) questionnaire. This tool includes five yes/no questions and people can complete it themselves or with assistance. The Spanish version of the SCOFF, validated for primary care use [41], sets a cutoff of two positive answers out of five. This cutoff has proven valid for detecting eating disorders in a primary care context [41], as an initial screening to identify those who may need further, more detailed evaluation. A previous study among Spanish adolescents reported a sensitivity of 73% (95% CI 63% to 83%), and a specificity of 78% (95% CI 75% to 80%) for detecting eating disorders [42].

### 2.2. Covariates

#### 2.2.1. Sociodemographic Factors

Participants reported their own sex and age. The Family Affluence Scale (FAS-III) [43] was used to determine their socioeconomic status, scoring from 0 to 13 points.

#### 2.2.2. Lifestyle Factors

The Spanish version of the Youth Activity Profile (YAP-S), a 15-item self-report questionnaire, captured physical activity and sedentary behavior [44]. Validated for Spanish youth, this tool categorizes activities into school, outside school, and sedentary habits, using a 5-point Likert scale to reflect the previous week’s activities [45]. For sleep duration, we asked participants to tell us their usual bedtime and wake-up times on weekdays and weekends. We then calculated their average daily sleep duration with the formula: [(weekday sleep duration × 5) + (weekend sleep duration × 2)]/7. A self-administered food frequency questionnaire (FFQ), validated for the Spanish population, was used to estimate energy intake.

#### 2.2.3. Anthropometric Measurements

We measured adolescents’ body weight on an electronic scale, with an accuracy of 0.1 kg (Tanita BC-545, Tokyo, Japan), with the participants wearing minimal clothing. We used a portable stadiometer for height, with an accuracy of 0.1 cm (Leicester Tanita HR 001, Tokyo, Japan). To calculate body mass index (BMI), we divided the weight in kilograms by the height in meters squared.

### 2.3. Statistical Analysis

To assess the normality of the variables, we employed visual techniques such as density and quantile–quantile plots, as well as the Shapiro–Wilk test. Therefore, for this study, we reported median and interquartile range (IQR) for the continuous variables, and count and percentage for the categorical variables. Binary logistic regression analyses were used to calculate odds ratios (OR) and 95% confidence intervals (CI) to estimate the likelihood of disordered eating as a function of family meal frequency and social eating behavior scale score. In addition, we computed the predictive probabilities of having disordered eating according to the number of family meals or the points in the social eating behavior scale. Models were adjusted for age (years), sex (boys or girls), socioeconomic status (FAS-III score), physical activity (YAP-S physical activity score), sedentary behavior (YAP-S sedentary behavior score), overall sleep duration (minutes), body mass index (kg/m^2^), and energy intake. We conducted all statistical analyses using the R statistical software (version 4.3.2) (R Core Team, Vienna, Austria) and RStudio (version 2023.09.1+494) (Posit, Boston, MA, USA). We considered a *p*-value < 0.05 as the threshold for statistical significance.

## 3. Results

Table 1 shows the descriptive data of the sample of adolescents examined. The median of weekly family meals and social eating behavior in adolescents were 14.0 (IQR = 6.0) and 9.8 (IQR = 1.8), respectively. Furthermore, 30.2% of the sample showed disordered eating behavior.

Table 2 and Table 3 display the unadjusted and adjusted odds ratios of disordered eating behavior in adolescents per one additional family meal or social eating behavior scale point, respectively. After adjusting for several covariates, for each additional family meal, the likelihood of having disordered eating behavior was lower (OR = 0.96; 95% CI 0.93 to 0.9997, *p* = 0.049). On the other hand, a lower likelihood of having disordered eating behavior was observed for each additional point in the social eating behavior scale (OR = 0.85; 95% CI 0.77 to 0.93, *p* = 0.001). Both adjusted and unadjusted results show a lower probability of having a disordered eating behavior for each additional family meal or point on the social eating behavior scale.

Figure 1 displays the predictive probabilities of having disordered eating behavior for each additional family meal or point on the social eating behavior scale. The likelihood of having disordered eating behavior was 0.7% lower for each additional family meal (95% CI 0.01% to 1.4%, *p* = 0.046). Furthermore, for each additional point in the social eating behavior scale, a lower probability of having disordered eating behavior was observed (3.2%; 95% CI 1.4% to 5.0%, *p* < 0.001). Both the adjusted and unadjusted results show an inverse association between each additional family meal or point on the social eating behavior scale and the probability of having disordered eating behavior.

## 4. Discussion

Although the association between family meal frequency and disordered eating behavior has been pointed out [28,30,46], to our knowledge, no previous study has analyzed the association between social eating behavior and disordered eating behavior in adolescents. Overall, our findings suggest that more family meals and a higher social eating behavior are related to lower odds of disordered eating behavior among adolescents. However, the estimates noted for these associations were low, indicating that additional demographic-, anthropometric-, or lifestyle-related variables might also contribute to increased disordered eating behavior in adolescents. In the case of family meals, our results are in line with previous studies in the scientific literature [28,30,46]. For instance, Haines et al. [46] observed that adolescent girls who regularly had dinner with their family members most days or every day of the week were less inclined to start engaging in purging behaviors, binge eating, and frequent dieting in the subsequent year. In adolescent boys, the results were also comparable in both direction and strength, but due to the lower prevalence of disordered eating behavior, the estimates were less accurate and did not reach statistical significance.

The mechanisms behind this potential protective effect of family meals against disordered eating behavior may be explained by several factors [39]. For instance, regular family meals can provide a structured environment offering adolescents concrete examples of portion control and balanced nutrition [28]. In this sense, family meals may not only facilitate the development of regular eating patterns but also avoid the chaotic eating patterns often associated with disordered eating behavior [26]. They could also encourage healthier food choices through direct observation and active participation in balanced meal preparation, modeling healthy eating behaviors [38]. Furthermore, these shared meals could act as a space for open communication, providing an opportunity to express concerns and reinforce positive feedback regarding eating habits, which can be especially beneficial during the adolescent years [47], as this period involves significant changes and the acquisition of habits that may persist into the future [38]. Moreover, the communal aspect of eating can foster mindfulness and attunement to internal cues of hunger and satiety, essential elements for cultivating a healthy relationship with food [48]. Likewise, the practice of sharing meals can establish a supportive network that enhances positive self-esteem [49]. Henceforth, family meals may serve as a counterbalance to the influence of social networks [34,50] and peer pressures [21] that frequently impact eating behaviors.

Although the positive impact of family meal participation on the prevention of disordered eating behavior in young individuals (especially in girls) has generally been highlighted, it has also been pointed out that the protective effect of family meals may not be equal in all cases and may be influenced by certain factors at the family level in certain situations [39]. It is possible that households where family meals contribute to conflict or negative interactions may experience a diminished protective effect, or even a harmful effect [51]. This fact underscores the complexity of family dynamics, suggesting that simply increasing the number of family meals, without addressing the quality of interpersonal relationships, may prove insufficient in preventing or reducing disordered eating behavior. Another possible reason could be attributed to developmental disparities between adolescents and young adults in their susceptibility to family influences [52]. It is also conceivable that some familiar or social factors, such as parents’ socioeconomic and educational level or peer relationships, may change the potential relationship between family meals and eating behaviors, by changing the adolescent perception of family relationships, and compliance with parental guidance [53]. Additionally, these disparities might reflect cultural differences in the perception of family meals [54].

In our study, we also identified a significant inverse association between social eating behavior and the presence of disordered eating behavior among adolescents, implying that social engagement during meals may function as a protective factor against the development of these behaviors. The communal aspect of dining, in this sense, could foster a sense of belonging [55] and act as a buffer against the isolation often associated with disordered eating behavior [56]. Consequently, social eating might encourage the sharing of diverse food experiences and normalize eating habits within a peer-supported setting, a factor particularly influential during adolescence [27]. Our results align with previous studies demonstrating that adolescents who enjoy meals with others are less likely to exhibit symptoms of food-related psychological distress [25,29]. These findings highlight the importance of the quality of social interactions during meals, emphasizing that it is not merely the act of eating with others but the positive nature of these interactions [57,58] that contributes to nutritional health in children and adolescents [59]. However, it is worth noting that our findings diverge from other studies. For instance, Herman [60] suggested that social eating may expose individuals to social facilitation of eating, and consequently, to overeating. These discrepancies may reflect that the context and nature of social interactions during meals could be crucial [60,61]. In this line, it has been argued that positive and supportive social eating contexts may be protective, whereas those that are competitive or judgmental could exacerbate the risk of disordered eating behavior [39,62].

This study has several limitations that should be mentioned. The cross-sectional design lacks the capacity to establish causal relationships. To ascertain whether an increased frequency of family meals and social eating behavior directly contributes to a decrease in disordered eating behavior among adolescents, longitudinal research is needed. Furthermore, the reliance on self-reported data introduces the potential for recall bias or social desirability bias, impacting the accuracy of reported frequencies of family meals and social eating behaviors. Likewise, disordered eating behavior is influenced by multiple factors, and even though our analyses account for several covariates, it is conceivable that other variables not addressed in this study could impact the observed results. Moreover, another limitation is that our results may not be generalizable to the rest of the regions in Spain, as well as to other countries. This is because of cultural differences regarding the habit of eating with family or in the company of others [63], which could impact the obtained results. Conversely, this study has some strengths that include its contribution of cross-sectional evidence regarding the role of these eating-related factors in disordered eating behavior within an understudied population (i.e., adolescents). Additionally, our adjustment for several covariates, including sociodemographic, lifestyle, and anthropometric variables, adds robustness to the results. Nevertheless, despite these adjustments, it is important to acknowledge that residual confounding cannot be entirely ruled out.

## 5. Conclusions

While disordered eating behavior is complex and can be shaped by various factors, both family meals and social eating behavior emerge as significant factors inversely associated with this condition among adolescents. Given the substantial prevalence of disordered eating behavior in the global adolescent population [10], it becomes imperative to acknowledge and strengthen the protective influence of these social and familial factors. Promoting regular engagement in family meals and fostering positive social eating experiences could serve as effective strategies in public health initiatives aimed at mitigating the incidence of disordered eating behavior among the young population. Encouraging open communication and shared mealtimes within families can create a supportive environment that fosters a healthy relationship with food. Simultaneously, interventions should aim to cultivate positive social eating experiences, emphasizing the importance of enjoyable and inclusive eating settings.

## Figures and Tables

**Figure 1 nutrients-16-00951-f001:**
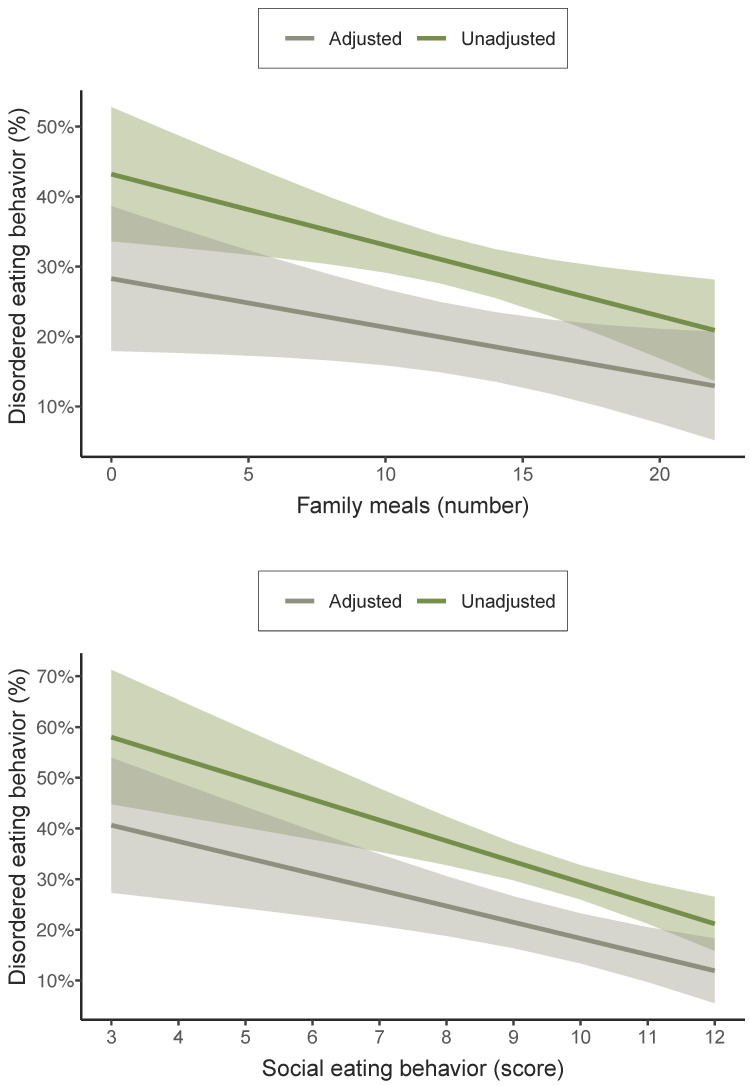
Predictive probabilities of having disordered eating behavior for each additional family meal or point on the social eating behavior scale among adolescents. Adjusted for age, sex, socioeconomic status, physical activity, sedentary behavior, overall sleep duration, body mass index, and energy intake.

**Table 1 nutrients-16-00951-t001:** Descriptive data of the study participants (*N* = 706).

Variables		Total
School	*IES Vicente Medina* (%)	376 (53.5)
	*IES Pedro Guillén* (%)	187 (26.5)
	*CE El Ope* (%)	143 (20.3)
Age (years)	Median (IQR)	14.0 (2.0)
Sex	Boys (%)	310 (43.9)
	Girls (%)	396 (56.1)
FAS-III (score)	Median (IQR)	8.0 (3.0)
YAP-S physical activity (score)	Median (IQR)	2.6 (0.9)
YAP-S sedentary behavior (score)	Median (IQR)	2.6 (0.8)
Overall sleep duration (minutes)	Median (IQR)	501.4 (71.8)
Body mass index	Median (IQR)	21.7 (6.0)
Energy intake (kcal)	Median (IQR)	2554.3 (1465.9)
Weekly family meals (number)	Median (IQR)	14.0 (6.0)
Social eating behavior (score) ^a^	Median (IQR)	9.8 (1.8)
SCOFF (score)	Median (IQR)	1.0 (2.0)
Disordered eating behavior ^b^	No (%)	493 (69.2)
	Yes (%)	213 (30.2)

Data are expressed as the median (interquartile range) or count (percentages). CE, *Cooperativa de Enseñanza*; FAS-III, Family Affluence Scale-III; IES, *Instituto de Educación Secundaria*; IQR, interquartile range; SCOFF, Sick, Control, One stone, Fat, Food; YAP-S, Spanish Youth Active Profile. ^a^ Social eating behavior scale ranges from 3 to 12 points^. b^ Cutoff point for eating disorders ≥ 2 points on the Sick, Control, One, Fat and Food (SCOFF) questionnaire.

**Table 2 nutrients-16-00951-t002:** Unadjusted and adjusted odds ratios of eating disorders in adolescents for each additional family meal in adolescents.

	Disordered Eating Behavior ^a^ (Outcome)	
Predictor	OR (95% CI, *p*-Value) Univariable	OR (95% CI, *p*-Value) Multivariable
Family meals global (per one meal)	0.95 (0.92 to 0.99, *p* = 0.005)	0.96 (0.93 to 0.9997, *p* = 0.049)
Sex		
Boys	Reference	Reference
Girls	2.32 (1.65 to 3.28, *p* < 0.001)	2.93 (2.00 to 4.34, *p* < 0.001)
Age (per one year)	1.03 (0.92 to 1.14, *p* = 0.620)	0.93 (0.82 to 1.05, *p* = 0.260)
FAS-III score (per one point)	0.92 (0.85 to 1.00, *p* = 0.040)	0.92 (0.85 to 1.00, *p* = 0.052)
YAP-S physical activity (per one point)	1.11 (0.87 to 1.41, *p* = 0.386)	1.27 (0.97 to 1.67, *p* = 0.085)
YAP-S sedentary behavior (per one point)	1.04 (0.80 to 1.35, *p* = 0.782)	1.03 (0.75 to 1.40, *p* = 0.859)
Overall sleep duration global (per one hour)	0.79 (0.66 to 0.95, *p* = 0.011)	0.86 (0.70 to 1.05, *p* = 0.144)
Body mass index (per one kg/m^2^)	1.12 (1.09 to 1.17, *p* < 0.001)	1.14 (1.10 to 1.19, *p* < 0.001)
Energy intake (per 1000 kcal)	1.04 (0.96 to 1.12, *p* = 0.280)	1.02 (0.94 to 1.12, *p* = 0.578)

^a^ Cutoff point for eating disorders ≥ 2 points on the Sick, Control, One, Fat and Food (SCOFF) questionnaire. CI, confidence interval; OR, odds ratio. FAS-III, Family Affluence Scale-III; YAP-S, Spanish Youth Active Profile.

**Table 3 nutrients-16-00951-t003:** Unadjusted and adjusted odds ratios of eating disorders in adolescents for each additional social eating behavior scale point in adolescents.

	Disordered Eating Behavior ^a^ (Outcome)	
Predictor	OR (95% CI, *p*-Value) Univariable	OR (95% CI, *p*-Value) Multivariable
Social eating behavior (per one point)	0.83 (0.76 to 0.90, *p* < 0.001)	0.85 (0.77 to 0.93, *p* = 0.001)
Sex		
Boys	Reference	Reference
Girls	2.32 (1.65 to 3.28, *p* < 0.001)	3.10 (2.11 to 4.61, *p* < 0.001)
Age (per one year)	1.03 (0.92 to 1.14, *p* = 0.620)	0.95 (0.84 to 1.07, *p* = 0.398)
FAS-III score (per one point)	0.92 (0.85 to 1.00, *p* = 0.040)	0.93 (0.86 to 1.02, *p* = 0.111)
YAP-S physical activity (per one point)	1.11 (0.87 to 1.41, *p* = 0.386)	1.28 (0.98 to 1.69, *p* = 0.076)
YAP-S sedentary behavior (per one point)	1.04 (0.80 to 1.35, *p* = 0.782)	1.00 (0.73 to 1.36, *p* = 0.995)
Overall sleep duration global (per one hour)	0.79 (0.66 to 0.95, *p* = 0.011)	0.86 (0.70 to 1.06, *p* = 0.148)
Body mass index (per one kg/m^2^)	1.12 (1.09 to 1.17, *p* < 0.001)	1.14 (1.10 to 1.19, *p* < 0.001)
Energy intake (per 1000 kcal)	1.04 (0.96 to 1.12, *p* = 0.280)	1.02 (0.94 to 1.11, *p* = 0.705)

^a^ Cutoff point for eating disorders ≥ 2 points on the Sick, Control, One, Fat and Food (SCOFF) questionnaire. CI, confidence interval; OR, odds ratio. FAS-III, Family Affluence Scale-III; YAP-S, Spanish Youth Active Profile.

## Data Availability

The data used in this review are available from the corresponding authors after reasonable request, since they pertain to minors, and privacy and confidentiality must be respected.

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
