# Peer review of "Family Meals and Social Eating Behavior and Their Association with Disordered Eating among Spanish Adolescents: The EHDLA Study"

_nutrients, 2024, doi:10.3390/nu16070951_

Round 1
Reviewer 1 Report
Comments and Suggestions for Authors
Lines 104-114 How did the authors choose school? What were characteristics of schools?
Lines 116-120 Why did the author exclude children who did not participate in physical class?
Author Response
Reviewer 1
Lines 104-114 How did the authors choose school? What were characteristics of schools?
Thank you for your comment. We selected all the school from this region (i.e., Valle de Ricote, Region of Murcia, Spain). The number of participants from each school has been added in Table 1. Thank you.
Lines 116-120 Why did the author exclude children who did not participate in physical class?
These participants were excluded because the majority of tests and questionnaires were administered during physical education classes. Thank you.
Reviewer 2 Report
Comments and Suggestions for Authors
Dear authors,
the manuscript presents an interesting line of research on the relationship between family meals, social eating behavior and the presence of disordered eating behaviors (DEB) in Spanish adolescents.
The research presents a observational study on the relationship between the presence of family meals and social eating behaviors with DEB in adolescents. It is important to clarify, as the authors comment, that this research should be carried out and complemented as a valuable public health intervention through a longitudinal research design. A important sample size of the study population should be taken into account.
Thus, the following recommendations for this study are to develop all the potential that could be extracted from this valuable sample of adolescents.
The title is recommended to indicate the region of the country where the research is conducted.
Thus, the abstract and introduction have the recommended length, as well as the revision of the bibliography is updated for the writing of the same.
The main and most important handicap of this manuscript is that the results expressed in tables and figures are scarce. Starting from 3 Variables and 3 Covariate, only 2 figures and 2 tables are presented, which are insufficient results to try to understand and discuss this research much more.
And thus to try to reach more profound conclusions.
Sincerely yours.
Comments on the Quality of English Language
No comments.
Author Response
Reviewer 2
Dear authors,
The manuscript presents an interesting line of research on the relationship between family meals, social eating behavior and the presence of disordered eating behaviors (DEB) in Spanish adolescents. The research presents a observational study on the relationship between the presence of family meals and social eating behaviors with DEB in adolescents. It is important to clarify, as the authors comment, that this research should be carried out and complemented as a valuable public health intervention through a longitudinal research design. A important sample size of the study population should be taken into account. Thus, the following recommendations for this study are to develop all the potential that could be extracted from this valuable sample of adolescents.
Thank you for your time and feedback.
The title is recommended to indicate the region of the country where the research is conducted.
We have added the country. Thank you.
Thus, the abstract and introduction have the recommended length, as well as the revision of the bibliography is updated for the writing of the same.
Thank you for your feedback.
The main and most important handicap of this manuscript is that the results expressed in tables and figures are scarce. Starting from 3 Variables and 3 Covariate, only 2 figures and 2 tables are presented, which are insufficient results to try to understand and discuss this research much more. And thus to try to reach more profound conclusions.
Following your indication, we have added the full results of the GLM models in the Supplementary Material. Furthermore, we have added some information about the potential confounders. Thank you.
Reviewer 3 Report
Comments and Suggestions for Authors
The manuscript under review studied 706 adolescents from a region in Spain to describe the relationship between family meals and social eating behavior on disordered eating behaviors (DEB) which can be a precursor to eating disorders. Overall the manuscript is very well-written and demonstrates that even after adjusting for socio-economic factors, both family meals and social eating have protective benefits on DEB.
I would like to suggest a few minor additions/changes to further strengthen the manuscript.
1. Lines 144-148: Providing the sensitivity/specificity for the cutoff of the SCOFF would be helpful to readers who may be unfamiliar with the questionnaire.
2. Figure 1: The colors used in the figure have a low-level of contrast. I would suggest changing the color scheme or incorporating different line types to delineate the lines better.
3. Lines 281-292: What about the limitation that the study was performed in a particular region of Spain? Is it generalizable to a larger population?
4. Line 72 & Line 115: Small grammatical errors in these two lines.
Author Response
Reviewer 3
The manuscript under review studied 706 adolescents from a region in Spain to describe the relationship between family meals and social eating behavior on disordered eating behaviors (DEB) which can be a precursor to eating disorders. Overall the manuscript is very well-written and demonstrates that even after adjusting for socio-economic factors, both family meals and social eating have protective benefits on DEB.
Thank you so much for your time and feedback.
I would like to suggest a few minor additions/changes to further strengthen the manuscript.
- Lines 144-148: Providing the sensitivity/specificity for the cutoff of the SCOFF would be helpful to readers who may be unfamiliar with the questionnaire.
The next information has been added: “A previous study among Spanish adolescents reported a sensitivity of 73% (95% CI 63% to 83%), and a specificity of 78% (95% CI 75% to 80%) for detecting eating disorders [42]”. Thank you.
- Figure 1: The colors used in the figure have a low-level of contrast. I would suggest changing the color scheme or incorporating different line types to delineate the lines better.
Thank you for your comment. The figures look worse because they have been inserted directly into the manuscript. In the PDF version, they have better contrast. If the editor requires it, we will gladly modify the colors/contrasts without any issue.
- Lines 281-292: What about the limitation that the study was performed in a particular region of Spain? Is it generalizable to a larger population?
We have added this limitation. Thank you.
- Line 72 & Line 115: Small grammatical errors in these two lines.
Done. Thank you.
Round 2
Reviewer 2 Report
Comments and Suggestions for Authors
Dear authors.
The new version of the manuscript mainly improves the results section. In particular the new 2 additional tables are appropiate. For this rason these new supplementary tables on the calculation of the ODS ratio should be included in the Results Section and not Annexed. Since this would give more power and credibility to the article for this section of Clinical Nutrition.
Sincerally yours.
Author Response
Done. Thank you for your suggestion.